# Training-Free Speedup for Retrieval-Augmented Generation with Staged Parallel Speculation

## Abstract

Retrieval-augmented generation (RAG) leverages external knowledge bases to enhance the quality of answers produced by large language models (LLMs). However, retrieving relevant documents from large-scale databases can be time-consuming, and existing RAG methods primarily focus on improving accuracy while often overlooking latency. In this paper, we introduce *Staged Parallel Speculation (SPS)*, a training-free RAG framework that achieves substantial latency reduction without sacrificing answer quality. Unlike prior approaches that rely on task-specific training or model modifications, SPS is a plug-and-play method that requires no changes to the underlying models. Our framework enables the inference and retrieval systems to run in parallel during staged retrieval, thereby eliminating frequent pauses in the inference process and significantly accelerating generation. Furthermore, at each retrieval-generation stage, SPS first uses a model to generate multiple candidate answer chunks in parallel and then selects the most reliable output based on self-consistency among the candidates, thereby further improving answer quality. Extensive experiments across multiple benchmark datasets show that SPS consistently surpasses training-free RAG baselines by achieving higher accuracy with at most 57% lower latency, while still reaching 96% of the performance of finetuning-based methods, making it a practical choice for deployment in latency-sensitive applications such as agentic systems, enterprise knowledge management, or healthcare support.

## 1 Introduction

Large language models (LLMs) have achieved remarkable success in natural language processing, excelling in tasks like text generation, translation, and question answering (Brown et al., 2020; Izacard & Grave, 2021). However, they face key challenges in knowledge-intensive problems. They are prone to factual errors, or hallucinations, where generated information appears plausible but is incorrect (Karpukhin et al., 2020). Furthermore, their reliance on static training data leads to knowledge obsolescence, especially in rapidly evolving domains (Gao et al., 2024).

Retrieval-Augmented Generation (RAG) has emerged as an effective paradigm to mitigate the limitations of traditional large language models in knowledge-intensive tasks by integrating external knowledge retrieval with language generation (Lewis et al., 2020). By retrieving relevant documents and incorporating them with the question as input, RAG significantly reduces factual inaccuracies and helps address knowledge obsolescence (Cheng et al., 2025). Most existing RAG research has primarily focused on improving answer accuracy (Xie et al., 2024; Feng et al., 2024; Su et al., 2024). These approaches often rely heavily on additional fine-tuning—either to enhance retrieval quality or to strengthen how LLMs leverage retrieved information. However, such tuning-based methods are costly, time-intensive, and risk reinforcing biases inherent in training data (Bommasani et al., 2022; Ravaut et al., 2025). Beyond this, RAG inherently involves retrieving large amounts of information, which must then be integrated into long prompts for inference (Ma et al., 2024; Ding et al., 2023). This process introduces substantial latency, yet prior work has paid little attention to efficiency challenges. The issue becomes especially pressing in agentic AI systems (Peng et al., 2025; Ruan et al., 2025), where autonomous decision-making and tool use are executed at scale: in such settings,

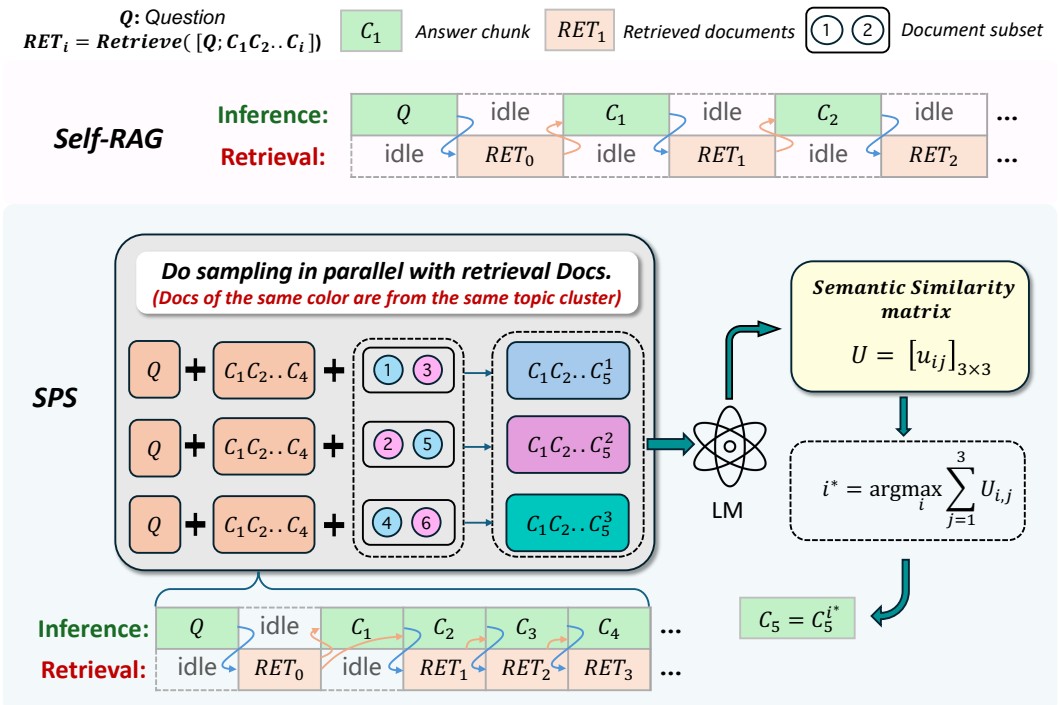

Figure 1: Comparison of Self-RAG and our proposed SPS. Unlike Self-RAG (Asai et al., 2024), which performs retrieval sequentially and incurs frequent idle time, SPS parallelizes retrieval with inference and employs multi-sampling with self-consistency selection, thereby reducing latency while maintaining answer quality.

reducing inference delay directly improves throughput and yields tangible economic benefits, making efficiency just as crucial as accuracy (Kim et al., 2025; Belcak et al., 2025).

In this work, we propose Staged Parallel Speculation (SPS), an efficient RAG framework that requires no additional training and significantly reduces generation latency without sacrificing answer accuracy. Our framework introduces staged retrieval. Instead of performing a single retrieval step before generation, as in traditional RAG methods, retrieval is periodically triggered during the generation process after a fixed number of tokens are generated. This approach ensures that the retrieved documents remain relevant to the latest context of the generation, allowing the model to better handle topic shifts that often occur when generating long answers to complex questions. Furthermore, at each retrieval–generation stage, we cluster the retrieved documents by content similarity and sample one representative from each cluster, which not only maximizes the coverage of diverse information but also reduces the input context length, thereby improving efficiency. We then generate candidate answer chunks in parallel based on different subsets. Finally, we select the best candidate answer chunk by leveraging self-consistency. Specifically, we identify the answer that is most semantically similar to all other candidates. Our contributions are summarized as follows:

- We propose Staged Parallel Speculation, a training-free RAG framework that substantially reduces generation latency without sacrificing accuracy.
- SPS parallelizes inference and retrieval through a staged mechanism that proactively and dynamically updates the retrieved documents in close alignment with the evolving generation context. Furthermore, it incorporates a self-consistency–based selection strategy that systematically identifies the most reliable candidate output at each retrieval–generation stage, leveraging semantic agreement among multiple candidates while remaining entirely training-free.
- Extensive experiments on multiple QA benchmarks show that SPS achieves substantial latency reduction while consistently outperforming all training-free baselines. In addition, SPS delivers performance comparable to, and in some cases exceeding, that of fine-tuning–based methods.

## 2 RELATED WORKS

### 2.1 RETRIEVAL AUGMENTED GENERATION

Retrieval-Augmented Generation (RAG) enhances the quality and relevance of generated outputs by retrieving external documents as context during generation, thereby reducing factual inaccuracies and limitations due to model knowledge staleness(Yu et al., 2025; Fan et al., 2024; Li et al., 2025; Gupta et al., 2024). RAG has demonstrated its versatility and effectiveness in grounding language models with timely and domain-specific information across a wide range of domains, including legal reasoning (Barron et al., 2025), medical decision support (Zhao et al., 2025), enterprise knowledge management (Packowski et al., 2024), and multimodal understanding (Choi et al., 2025). To further enhance the effectiveness of RAG, a growing line of work explores how to tune or train LLM to better integrate retrieval into the generation process. For instance, Self-RAG (Asai et al., 2024) introduces special reflection tokens and trains the model to decide when additional retrieval is necessary, thereby improving factual accuracy. SAIL (Luo et al., 2023) fine-tunes a pre-trained LLM on large-scale web search data to help it better filter out irrelevant content and prioritize useful evidence. Corrective RAG (Yan et al., 2024) adopts a lightweight retrieval evaluator trained to refine the quality of retrieved documents. Similarly, Toolformer (Schick et al., 2023) trains LLMs to call external APIs, such as search engines, at appropriate times to improve retrieval quality. While these approaches demonstrate significant gains in accuracy, they often focus primarily on correctness, overlooking the equally critical dimension of system efficiency, such as latency and computational overhead.

Accelerating RAG frameworks has become another active line of research. Jiang et al. (2024) proposed PipeRAG, which parallelizes the retrieval and generation modules to reduce latency by prefetching with stale queries and supporting flexible retrieval intervals. However, PipeRAG is built on top of the RETRO encoder–decoder framework (Borgeaud et al., 2021), which relies on a jointly trained encoder–decoder transformer to enhance retrieval and generation performance, rather than an LLM-based RAG pipeline. Consequently, PipeRAG requires additional training to construct its retrieval module and cannot leverage the strong zero-shot and in-context capabilities of modern decode-only large language models for RAG question-answering. Wang et al. (2024) introduced Speculative RAG, which improves generation efficiency by training a smaller model to first draft candidate outputs that are then verified by a larger model. Nevertheless, these approaches either are not LLM-based RAG methods or rely on task-specific training, which demands substantial time and computational resources and may further introduce data bias into the underlying models (Wei et al., 2025). Unlike prior methods that require task-specific training, SPS is plug-and-play and leaves underlying models frozen. During staged retrieval, it runs retrieval and inference in parallel, avoiding frequent inference stalls and substantially speeding up generation.

### 2.2 SPECULATIVE DECODING

The goal of Speculative Decoding (Stern et al., 2018; Leviathan et al., 2023; Chen et al., 2023; Xia et al., 2024) is to accelerate LLM inference through a draft-and-verify process. It first uses a smaller draft model to generate multiple future tokens, followed by a larger verification model that evaluates these draft tokens in parallel. The draft model can either be a smaller version from the same series as the verification model (Leviathan et al., 2023; Chen et al., 2023) or identical to the verification model (Zhang et al., 2023; Cai et al., 2024). Our method extends the draft-and-verify paradigm from token-level drafting to text chunk-level drafting. Additionally, we design a verification strategy tailored to our system, which leverages the idea of self-consistency to verify candidate answers based solely on their semantic properties, without relying on a larger verification model.

## 3 OUR APPROACH: STAGED PARALLEL SPECULATION

**Problem Setup.** In Staged Parallel Speculation, each entry can be represented as $(Q, \mathcal{D}, R, C, A)$. Given $Q$, a question or statement that requires additional knowledge to get the correct answer $A$; $\mathcal{D}$ is a set of documents. Let $C_t = (c_1, c_2, \ldots, c_t)$ denote the chunk sequence of the answer. At the $i$-th retrieval-generation step during inference, (i) language model $\mathcal{M}$ generate the current answer chunk $c_i$, based on the previous retrieved documents $R_{i-2} \in \mathcal{D}$ and the answer hisrtory $C_{i-1}$; (ii) retrieval model $\mathcal{R}$ retrieve related $n$ document $R_{i-1} = (r_1, r_2, \ldots, r_n) \in \mathcal{D}$ based on question $Q$ and all

---

**Algorithm 1:** Staged Parallel Speculation

1  **Data description:** Question $Q$, answer seq. $C_t = (c_1, \ldots, c_t)$, retrieved docs $R = (r_1, \ldots, r_n)$, number of subsets $m$, number of docs per subset $k$, retrieval model $\mathcal{R}$, generation model $\mathcal{M}$

2  **Result:** Completed answer seq. $C$

3  **Initialize:** $C \leftarrow \varnothing, R \leftarrow \varnothing$

4  **repeat**

5    $R \leftarrow \mathcal{R}(Q, C)$                   ▷ Retrieved in parallel by the retrieval system during inference.

6    Cluster the $n$ retrieved docs into $k$ groups:

$$\{g_1, \ldots, g_k\} \leftarrow R(r_1, \ldots, r_n)$$

    **repeat**

7      $s_i = \{\}$                                     ▷ $s_i$ is a subset

8      **for** $g_i$ *in* $\{g_1, \ldots, g_k\}$ **do**

9        $s_i \leftarrow s_i \cup \{\text{random.sample}(g_i)\}$   ▷ Sample one document from each group into subset $s_i$

10    **until** *$m$ subsets $\{s_1, \ldots, s_m\}$ are generated*

11    **for** $s_j \in \{s_1, \ldots, s_m\}$ *(in parallel)* **do**

12      $c_{t+1}^j \leftarrow \mathcal{M}.\text{generate}(C, s_j)$           ▷ Generate $m$ candidate chunks in parallel

13    **Append** $c_{t+1}^j$ to $C$ to form candidate answers $\{[C, c_{t+1}^j] \mid 1 \leq j \leq m\}$

14    $U_{i,j} = \text{sim}_{\cos}(\text{emb}[C; c_{t+1}^i], \text{emb}[C; c_{t+1}^j]), \quad U \in \mathbb{R}^{m \times m}$

                                           ▷ Use embeddings to compute similarity matrix

15    **for** $i = 1$ **to** $m$ **do**

16      $\mu_i \leftarrow \sum_{j=1}^{m} U_{i,j}$                ▷ Average similarity for $i$-th candidate answer

17    $j^* = \arg\max_j \mu_j$

18    $C \leftarrow [C, c_{t+1}^j]$             ▷ Select best candidate answer and continue next round

19  **until** *EOS token in $C$*

---

answer history $C_{i-1}$; (iii) after $T$ rounds, we have a final answer $C_T$. The primary objective is to ensure that $C_T$ is factually consistent with the ground-truth answer $A$, while significantly improving the efficiency of the generation process.

### 3.1 OVERVIEW

We propose Staged Parallel Speculation, a RAG framework that focuses on improving efficiency under a training-free setting. As illustrated in Figure 1, this framework enables the inference system and retrieval system to operate independently during staged retrieval. At each retrieval-generation stage, after generating multiple draft chunks, we append each new chunk to the current answer sequence to form a set of complete draft answers. We then apply self-consistency by selecting the draft answer that is most semantically similar to the others and use it as the updated answer sequence for the next generation step. This process is repeated iteratively until the final answer is fully generated.

Specifically, as compactly described in Algorithm 1, for a question $Q$ that has not yet started generation, its answer sequence $C$ and retrieval document set $R$ are initialized as empty (**Line 3**). The main loop then begins (**Line 4**). The retrieval process runs concurrently with the inference process, continuously retrieving relevant documents from the database using a query formed by concatenating the original question with the current answer sequence. If the answer sequence is empty, the retrieval is performed using only the question. Since no answer sequence $C$ exists at this point, we first perform an initial retrieval using the question $Q$ to obtain the corresponding retrieval documents $R$ (**Line 5**). The retrieved document set $R$ is then clustered into $k$ groups using K-means clustering, and one document is sampled from each cluster to form a subset. This ensures that the sampled documents capture diverse perspectives from the retrieved content. The $i$-th subset is denoted as $s_i$

**(Line 6 to 10)**. By processing the documents in this way, the subset better captures the diversity and multiple perspectives of the retrieved content.

Then, each subset, together with the generated answer sequence $C$, is assigned to an LLM to independently and concurrently generate the next chunk of the answer sequence, $c_{t+1}$ **(Line 12)**. Generating candidate chunks in parallel is essential to ensure improved efficiency. Next, we concatenate each of the $m$ generated draft chunks to the current answer sequence to form $m$ draft answers **(Line 13)**. We then compute a pairwise similarity matrix among the draft answers using an embedding model **(Line 14)**. For each draft answer, we calculate a similarity score by summing its corresponding row (or column) in the matrix **(Line 15)**. The draft answer with the highest similarity score is selected as the updated answer sequence **(Line 17)**, which is then used in the next generation step. This process is repeated until the $EOS\ token$ is detected in the answer sequence $C$, indicating that the answer generation for the question $Q$ is complete.

We next detail the design of SPS component by component: we begin with multi-perspective sampling of retrieved evidence, then describe the stagewise parallel retrieval–inference architecture, and finally present the self-consistency–based selection mechanism

## 3.2 Multi-Perspective Sampling

For constructing diverse subsets from the retrieved documents, we follow the approach proposed by Wang et al. (2024). Specifically, we first use an open-source embedding model to compute embedding vectors for the textual content of all retrieved documents.

$$\text{emb}(r_1), \ldots, \text{emb}(r_n) = \mathcal{E}(r_1, \ldots, r_n) \tag{1}$$

where $\mathcal{E}$ is an open-source embedding model that embeds the document content into a vector representation; $\text{emb}(r_i)$ is the embedding for the retrieved document $r_i$. After obtaining the embeddings, we apply K-means clustering to divide the documents into $k$ groups. We then sample one document from each group into a document subset $s$ so each subset contains $k$ documents of diverse contents. In total, we construct $m$ subsets for parallel inference.

$$\{\mathbf{g}_1, \ldots, \mathbf{g}_k\} = \textbf{K-Means}(\text{emb}(r_1), \ldots, \text{emb}(r_n)) \tag{2}$$

$$s = \left\{ \texttt{random.sample}(g) \,\middle|\, g \in \{\mathbf{g}_i\}_1^k \right\} \tag{3}$$

The retrieved documents may contain diverse content due to the ambiguity inherent in the query. Constructing subsets in this manner helps reduce redundancy and promote diversity among the documents used for answer draft generation.

## 3.3 Parallelism

PipeRAG (Jiang et al., 2024) parallelizes the retrieval and generation modules to reduce latency. SPS advances the paradigm with a stagewise dense retrieval architecture that conditions each retrieval step on the entire question plus the full answer history, not just the latest chunk. This design captures substantially richer semantics, yields more relevant evidence, and improves answer reliability without any dual-encoder training or model modifications. In short, SPS is a plug-and-play framework that unifies parallel execution with semantically complete, history-aware retrieval.

As illustrated in Figure 1, before the answer sequence is generated, the retrieval system first uses the question as the initial retrieval query, and the retrieved documents are denoted as $RET_0$. $RET_0$ is then used to generate the first two answer chunks, $C_1$ and $C_2$. While the inference system is generating chunk $C_1$, the retrieval system remains idle. During the generation of chunk $C_2$, the retrieval system runs in parallel to retrieve documents based on the current context $[Q; C_1]$, resulting in $RET_1$. $RET_1$ is subsequently used for generating chunk $C_3$, while the retrieval system concurrently performs retrieval for $RET_2$ during the generation of chunk $C_3$. This process continues iteratively until the entire answer sequence is generated.

In Self-RAG, the inference and retrieval systems operate sequentially: retrieval is performed only after each chunk is generated and only if needed. The retrieved documents are based on the most recent context – for example, the generation of chunk $C_3$ uses $RET_2$, which is retrieved based on the complete context up to $C_2$. In contrast, our system uses $RET_1$ for generating $C_3$, meaning it is based

on a slightly earlier context. This introduces a one-chunk delay in the retrieval context. However, this trade-off enables full parallelization between inference and retrieval processes, thereby significantly reducing the overall end-to-end latency.

### 3.4 Self-consistency Selection

We leverage self-consistency to evaluate draft answers. Specifically, after generating $m$ draft chunks, we append each of them to the current answer sequence to form $m$ draft answers $\{[C, c_{t+1}^j] \mid 1 \leq j \leq m\}$. We then encode these draft answers using an embedding model and compute a cosine similarity matrix:

$$U_{i,j} = \text{sim}_{\cos}(\text{emb}[C; c_{t+1}^i], \ \text{emb}[C; c_{t+1}^j]), \quad U \in \mathbb{R}^{m \times m} \tag{4}$$

The similarity score for each draft answer is obtained by summing its corresponding row (or column) in the matrix: $\mu_i \leftarrow \sum_{j=1}^m U_{i,j}$, then the draft answer with the highest similarity score is selected to update the answer sequence for the next generation step:

$$j^* = \arg\max_j \mu_j, \ C \leftarrow [C, c_{t+1}^{j^*}] \tag{5}$$

In our case, the additional overhead comes from using an embedding model to encode the answer sequences and compute cosine similarity scores between them. Compared to Speculative RAG, which generates additional rationales and relies on a large verifier model, our approach is substantially more efficient, as the embedding model is significantly smaller, leading to lower memory usage and reduced runtime.

## 4 Experiments

### 4.1 Tasks and Datasets

We evaluate our proposed SPS framework on five retrieval-augmented generation benchmarks: PubHealth (Zhang et al., 2023), ARC-Challenge (Clark et al., 2018), TriviaQA-unfiltered (Joshi et al., 2017), PopQA (Mallen et al., 2023), and ALCE-ASQA (Gao et al., 2023). These datasets span a diverse range of knowledge-intensive tasks, including short-form, long-form, and closed-set question answering. All experiments are conducted in a zero-shot setting, where models receive task instructions without access to any in-context examples (Wei et al., 2022). Detailed evaluation setups, including test-time prompts and metrics, are provided in Appendix A.

**Closed-set QA.** PubHealth and ARC-Challenge are closed-set QA tasks. PubHealth is a fact verification dataset requiring models to classify claims as supported or not based on medical evidence. ARC-Challenge is a multiple-choice science QA benchmark sourced from standardized tests. For both datasets, we use accuracy as the primary metric to assess whether the model output matches the ground-truth label.

**Short-form QA.** TriviaQA and PopQA are open-domain QA benchmarks that require models to answer factual questions using retrieved documents. TriviaQA consists of naturally occurring questions from trivia sources, while PopQA emphasizes rare-entity coverage by focusing on long-tail Wikipedia questions. For both datasets, we evaluate whether the generated answer includes the gold reference answer, following Mallen et al. (2023); Schick et al. (2023).

**Long-form QA.** ALCE-ASQA is a long-form question answering benchmark where models are required to produce multi-sentence answers grounded in retrieved evidence. We adopt standard metrics from prior work to evaluate different aspects of model performance, including correctness (string exact match and ROUGE-L (Lin, 2004), fluency (via MAUVE (Pillutla et al., 2021)).

### 4.2 Baselines

**Standard RAG** For standard RAG, we include all retrieved documents in the prompt as contextual input. Detailed prompt templates are provided in Appendix A. We conduct standard RAG experiments

using pretrained LLMs including Mistral$_{7B}$, Mistral-Instruct$_{7B}$ (Jiang et al., 2023), Alpaca$_{7B,13B}$ (Dubois et al., 2024). For the instruction-tuned LMs, we use the official system prompt or instruction format used during training if available. We also include the performance of Toolformer (Schick et al., 2023) and SAIL (Luo et al., 2023), which are originally reported from Asai et al. (2024). Toolformer$_{6B}$ is an LM instruction-tuned to use tools, including a search engine, and SAIL$_{7B}$ is an LM instruction-tuned on the Alpaca instruction tuning set augmented with search results from different source, such as Wikipedia.

**Self-Reflective RAG and Corrective RAG** Self-Reflective RAG (Self-RAG) (Asai et al., 2024) and Corrective RAG (CRAG) (Yan et al., 2024) represent more advanced variants of the RAG framework that aim to enhance the quality of contextual information obtained through retrieval. CRAG incorporates an external evaluator to assess and refine the initially retrieved documents prior to answer generation. In contrast, Self-RAG leverages instruction tuning to enable the language model to produce special self-reflection tags. These tags prompt the model to dynamically retrieve additional documents when needed and to critically evaluate the relevance of retrieved content before generating a response. Self-CRAG integrates the self-reflective mechanism of Self-RAG with CRAG by applying it to CRAG's refined retrievals. For comparison, we directly include the performance of Self-RAG, CRAG, and Self-CRAG based on Mistral$_{7B}$ as reported in Wang et al. (2024).

**Pipe-Style RAG** PipeRAG (Jiang et al., 2024)achieves parallelization of retrieval and generation on top of the RETRO (Borgeaud et al., 2021) framework, and further improves generation quality while reducing latency by allowing flexible retrieval intervals. However, this approach relies on a jointly trained encoder–decoder transformer architecture and is therefore not directly compatible with decode-only LLMs. To enable a fairer comparison, we implement a Pipe-style RAG variant that is LLM-based. Specifically, we use the same retriever as other baselines to perform staged retrieval and employ an LLM to generate each chunk, repeating this process until a complete answer is produced.

**Speculative RAG** Speculative RAG (Wang et al., 2024) is a recent state-of-the-art RAG framework that incorporates the idea of speculative decoding to accelerate answer generation. It employs a small draft model to generate multiple candidate answers, which are then verified by a larger model to select the final output. This two-stage approach improves generation efficiency while maintaining strong answer quality. We include Speculative RAG as a competitive baseline in our comparison.[1]

### 4.3 Experimental settings

To ensure fair comparison, we conduct all experiments using LLMs with approximately 7 billion parameters. Specifically, we use open-source models Alpaca$_{7B}$, Mistral$_{7B}$ (v0.1), and Mistral-Instruct$_{7B}$ (v0.1) for answer generation, without applying any additional fine-tuning or modifications. All models are evaluated in a zero-shot setting, where only task instructions are provided, without any in-context demonstrations. To encode both the retrieved documents and the draft answers, we adopt a lightweight embedding model, `bge-large-en-v1.5` (Xiao et al., 2023). Inference is performed using greedy decoding (temperature = 0), and we set the chunk size to 50 tokens by default for all experiments. For consistency with Wang et al. (2024) and Asai et al. (2024), we use **Contriever-MS MARCO** (Izacard et al., 2022) as our retrieval model. For TriviaQA, PopQA, PubHealth, and ARC-Challenge datasets, we follow the same retrieval configuration as Wang et al. (2024), retrieving the top-10 documents for each question. We set $k = 5$ and $m = 5$ to create 5 document subsets, each containing 5 documents. This subset construction helps ensure broad coverage of the retrieved evidence across candidates. For the ALCE-ASQA dataset, we adopt the same retrieval setup as Asai et al. (2024), retrieving the top-5 documents and using $k = 3$ and $m = 5$ to create 5 document subsets, each containing 3 documents.

### 4.4 Performance Analysis

**Comparison with training-free baselines.** Table 1 shows that SPS consistently outperforms standard RAG across different backbones and tasks. Notably, when applied to Alpaca-7B, SPS

---

[1]For the fine-tuning-based Speculative RAG, we directly report the experimental results from the original paper. In addition, we report a training-free variant of Speculative RAG as a baseline for fair comparison.

Table 1: Overall experiment results on five tasks. * indicates concurrent or recent results reported by concurrent work. - indicates numbers that are not reported by the original papers or are not applicable. em, rg, mau, denote str-em, ROUGE (correctness); MAUVE (fluency).

| | Short-form | | Closed-set | | Long-form generations | | |
|---|---|---|---|---|---|---|---|
| | PopQA | TQA | Pub | ARC | ASQA | | |
| RAG Method | (acc) | (acc) | (acc) | (acc) | (em) | (rg) | (mau) |
| **Fine-tuning based methods** | | | | | | | |
| *Standard RAG* | | | | | | | |
| Toolformer*$_{6B}$ (Schick et al., 2023) | - | 48.8 | - | - | - | - | - |
| SAIL*$_{7B}$ (Luo et al., 2023) | - | - | 69.2 | 48.4 | - | - | - |
| *Self-Reflective RAG & Corrective RAG* | | | | | | | |
| Self-RAG*$_{Mistral-7B}$ (Asai et al., 2024) | 52.68 | 64.84 | 72.44 | 74.91 | - | - | - |
| CRAG*$_{Mistral-7B}$ Yan et al. (2024) | 49.46 | 59.03 | 59.04 | 74.84 | - | - | - |
| Self-CRAG*$_{Mistral-7B}$ Yan et al. (2024) | 56.11 | 65.43 | 72.85 | 75.26 | - | - | - |
| *Speculative RAG (Wang et al., 2024)* | | | | | | | |
| $\mathcal{M}_{Verifier-Mistral-7B} + \mathcal{M}_{Drafter-7B}$* | 56.75 | 73.91 | 75.79 | 76.19 | - | - | - |
| $\mathcal{M}_{Verifier-Mixtral-8\times7B} + \mathcal{M}_{Drafter-7B}$* | 57.54 | 74.24 | 76.60 | 80.55 | - | - | - |
| **Training free methods** | | | | | | | |
| *Standard RAG* | | | | | | | |
| Alpaca$_{7B}$ (Dubois et al., 2024) | 40.96 | 43.56 | 38.94 | 44.39 | 24.97 | 23.11 | 25.24 |
| Alpaca$_{13B}$ (Dubois et al., 2024) | 46.10 | 64.38 | 55.34 | 57.63 | 26.62 | 24.34 | 33.83 |
| Mistral$_{7B}$ (Jiang et al., 2023) | 32.59 | 53.50 | 35.26 | 43.51 | 25.42 | 23.87 | 44.76 |
| Mistral-Instruct$_{7B}$ (Jiang et al., 2023) | 42.03 | 65.83 | 43.16 | 47.61 | 26.19 | 24.92 | 46.41 |
| *Pipe-Style RAG (Jiang et al., 2024)* | | | | | | | |
| Pipe-Style RAG$_{Mistral-7B}$ | 32.47 | 54.16 | 34.19 | 41.28 | 25.23 | 24.71 | 42.37 |
| *Speculative RAG (Wang et al., 2024)* | | | | | | | |
| $\mathcal{M}_{Verifier-Mistral-7B} + \mathcal{M}_{Mistral-7B}$ | 34.63 | 57.51 | 38.16 | 47.38 | 25.07 | 25.74 | 54.12 |
| $\mathcal{M}_{Verifier-Mixtral-8\times7B} + \mathcal{M}_{Mistral-7B}$ | 39.37 | 60.49 | 42.52 | 49.31 | 25.49 | 27.94 | 56.72 |
| *SPS (Ours)* | | | | | | | |
| SPS$_{Alpaca-7B}$ | 48.39 | 67.87 | 59.17 | 65.96 | 28.38 | 34.30 | 61.79 |
| SPS$_{Mistral-7B}$ | 50.11 | 68.12 | 64.35 | 71.76 | 26.84 | 31.52 | 63.81 |

surpasses standard RAG built on Alpaca-13B, on every benchmark, e.g., +3.49% on TriviaQA and +8.33% on ARC-Challenge, despite using a smaller model. Compared with the Pipe-style RAG baseline, SPS consistently achieves higher accuracy (up to 30.48% improvement on the ARC dataset). Against *training-free* Speculative RAG, SPS$_{Mistral-7B}$ delivers sizeable gains (e.g., +10.74% on PopQA vs. a Mixtral-8×7B verifier with a Mistral-7B drafter), and improves ASQA correctness/fluency as well. We attribute this to SPS's training-free, self-consistency selection over parallel drafts: when no tuning is allowed, a learned drafter/verifier pair is less reliable, whereas SPS directly exploits semantic agreement among candidates without relying on trained rationale quality.

**Comparison with fine-tuning-based baselines.** As illustrated in Table 1, even when compared against strong fine-tuned baselines, SPS achieves comparable or superior performance. For instance, under the same backbone model Mistral-7B, SPS attains higher accuracy than CRAG (+8.84%), SelfRAG (+3.28%), and Self-CRAG (+2.69%) on TriviaQA. Its accuracy is only slightly below that of Speculative RAG with a fine-tuned 7B drafter and a Mistral-7B verifier ($-5.79\%$). These results highlight that SPS not only closes the gap with fine-tuning-based approaches but, in some cases, surpasses them, providing a practical alternative that eliminates the cost and complexity of additional training while retaining high answer quality.

## 4.5 LATENCY ANALYSIS

We conduct our latency analysis experiments on a server equipped with an AMD Ryzen Threadripper PRO 5975WX CPU and four NVIDIA RTX 6000 Ada GPUs. Specifically, we compare the inference latency of Standard RAG, Self-RAG, Speculative RAG, and our proposed SPS method across five datasets: PopQA, TriviaQA, PubHealth, ARC-Challenge, and ALCE-ASQA. Following the evaluation protocol of Wang et al. (2024), we randomly sample 100 examples from each dataset and report the average latency. For Standard RAG, we use Mistral$_{7B}$ as the generation model. Self-RAG

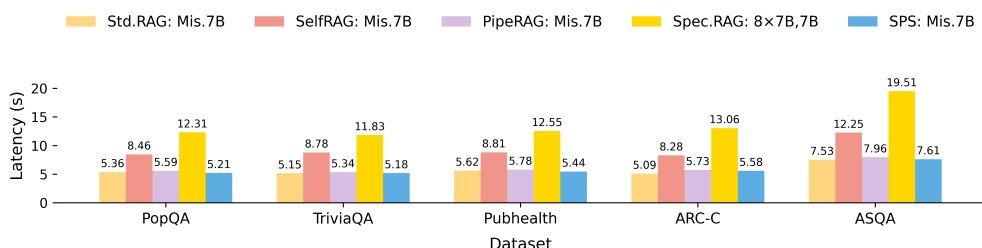

Figure 2: Latency analysis of Standard RAG, Self-RAG, Pipe-Style RAG, Speculative RAG(using Mixtral$_{8\times7B}$ as verifier and Mistral$_{7B}$ as drafter) and SPS on PopQA, TriviaQA, PubHealth, ARC-Challenge and ALCE-ASQA. The latency varies across different datasets due to different retrieved document lengths. SPS achieves lower latency than all other baselines across all evaluated datasets.

is evaluated using Mistral$_{7B}$, while Speculative RAG adopts the configuration reported as its best-performing setting—using Mistral$_{7B}$ as the drafter and Mixtral$_{8\times7B}$ as the verifier. Our SPS method is evaluated with Mistral$_{7B}$. For Standard RAG and Self-RAG, we retrieve the top-10 documents and include all of them directly in the prompt. For Speculative RAG, we retrieve the top-10 documents and set $k = 5$, $m = 5$. For our SPS method, we also retrieve the top-10 documents and adopt $k = 5$, $m = 5$, consistent with our previous experimental setup. For SPS and Speculative RAG, we launch 5 endpoints of the generation model to perform parallel drafting across document subsets on five datasets.

To simulate realistic deployment scenarios, we set the batch size to 1 (i.e., processing one query at a time). We also include document retrieval time in our measurements by performing retrieval immediately prior to generation, reflecting real-world system behavior. As shown in Figure 2, SPS achieves the lowest latency across all five datasets. This efficiency can be attributed to the parallel design of the retrieval and inference systems, as well as the fact that SPS does not require a larger model for verification.

### 4.6 COMPREHENSIVE COMPARISON

We present a comprehensive comparison of different RAG methods across four key dimensions: training requirement, GPU memory usage, inference latency, and answer accuracy. We adopt the same model configurations as used in the latency analysis. For GPU memory usage, although certain implementations of Mixtral-$8\times7B$ (e.g., with vLLM) support memory optimizations such as lazy loading, we measure memory under a standardized setting for fair comparison: loading models in FP16 using HuggingFace Transformers with a batch size of 1 during inference. For latency and accuracy, we report the average values across four datasets shared by all methods: TriviaQA, PopQA, PubHealth, and ARC-Challenge. As shown in Table 2, our SPS method achieves a strong balance among all evaluated dimensions. Without any additional training, our SPS improves performance by **54.3%** over Standard RAG while maintaining nearly the same latency. Compared with the training-free Speculative RAG, SPS achieves a **57%** reduction in latency together with a **32%** accuracy gain. Moreover, our training-free approach reaches **96%** of the performance of the fine-tuning-based Self-RAG and **88%** of the fine-tuning-based Speculative RAG.

## 5 CONCLUSION

This work introduces SPS, an efficient, training-free retrieval-augmented generation (RAG) framework that balances answer accuracy and inference latency. By combining staged retrieval, parallel draft generation, and self-consistency-based selection, SPS maintains competitive performance while significantly reducing computational overhead. Extensive experiments show that SPS substantially outperforms existing training-free baselines, achieving up to 57% lower latency and 32% higher accuracy. Furthermore, SPS attains performance comparable to fine-tuning-based methods, demon-

Table 2: Comprehensive Comparison of different RAG Methods.

| RAG Method | Training Required | Latency (s) | acc (%) |
|---|---|---|---|
| Self-RAG$_{\text{Mistral-7B}}$ | Yes | 8.58 | 66.22 |
| Spec.RAG$_{\text{Mixtral-8×7B,Drafter-7B}}$ | Yes | - | 72.23 |
| Spec.RAG$_{\text{Mixtral-8×7B,Mistral-7B}}$ | No | 12.44 | 47.92 |
| Standard RAG$_{\text{Mistral-7B}}$ | No | 5.31 | 41.21 |
| Pipe-Style RAG$_{\text{Mistral-7B}}$ | No | 5.61 | 40.53 |
| SPS$_{\text{Mistral-7B}}$ | No | 5.35 | 63.59 |

strating that high-quality RAG can be achieved without additional training. Overall, SPS offers a new paradigm for building RAG systems that break free from training dependency while improving accuracy and efficiency, making it especially suitable for real-world deployment under strict latency and resource constraints.

## ETHICS STATEMENT

This work adheres to the ethical standards of the ICLR community. Our research does not involve human subjects, sensitive personal data, or any information that could compromise privacy or security. All datasets used are publicly available and widely adopted in the research community. We believe the methods and results presented contribute positively to the advancement of AI research and do not foresee potential misuse beyond common considerations in language model research.

## REPRODUCIBILITY STATEMENT

We are committed to ensuring the reproducibility of our work. All datasets, preprocessing steps, model configurations, and evaluation protocols are described in detail in the main paper and appendix. We also provide source code, scripts, and instructions to replicate our experiments, which will be made publicly available upon publication.

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

## A  EXPERIMENTAL DETAILS

### A.1  RETRIEVAL SETUP DETAILS.

To ensure fair comparison in document retrieval, we use the dataset provided by the official Self-RAG implementation. This dataset includes questions along with their top-20 retrieved documents, obtained via a hybrid retriever combining Contriever-MS MARCO[2] and Google Programmable Search. Speculative RAG adopts the same setup, allowing our evaluation to eliminate variability introduced by differences in retrieved content. For our latency experiments, we also use Contriever-MS MARCO as the retriever, along with the 2018 Wikipedia dump (over 21 million passages) as the retrieval corpus—identical to the configuration used in Self-RAG.[3] This setup more closely reflects realistic retrieval-augmented generation scenarios. In our environment, we observe that each retrieval query takes approximately 2.5 to 3.5 seconds on average.

### A.2  DETAILED EXPERIMENTAL SETTINGS FOR INDIVIDUAL DATASETS.

All datasets used in this work are preprocessed versions provided by the open-source project released by Asai et al. (2024). For TriviaQA, PopQA, PubHealth, and ARC-Challenge, we use accuracy as the primary evaluation metric. Specifically, we determine whether the gold answer appears in the model-generated output.

For TriviaQA and PopQA, the official datasets provide multiple answer variants that account for case and surface form differences. Therefore, we apply direct string matching against these variants.

In contrast, PubHealth and ARC-Challenge require classification-style outputs. PubHealth involves assessing whether the retrieved documents support a given health claim, while ARC-Challenge consists of multiple-choice science questions. For both tasks, we provide explicit prompts instructing the model to output a specific label or option. In practice, we observe that models from the Mistral-Instruct family follow the instructions reliably and generate outputs in the expected format. However, models from the Alpaca family often produce overly concise responses and frequently fail to follow instructions (e.g., omitting the option label when asked to provide both the label and the option content), making direct evaluation difficult. To address this issue, we use **Gemini 2.0 Flash** to automatically evaluate outputs from Alpaca models. We provide Gemini with the input question, answer choices, gold answer and label, and the model's response, and ask it to assess whether the prediction is correct.

For ALCE-ASQA, we adopt the official evaluation script[4] released with the dataset, which computes exact match, ROUGE, and MAUVE scores based on the generated long-form answers.

### A.3  TASK-SPECIFIC INSTRUCTIONS.

Table 6 shows the list of instructions used on different datasets during evaluations.

## B  MORE EXPERIMENTAL RESULTS

### B.1  ABLATION STUDIES

We conduct ablation studies on ASQA to evaluate the contributions of each component in our SPS framework. Specifically, we examine (1) staged retrieval, (2) self-consistency selection, and (3) multi-perspective sampling. As shown in Table 3, removing any of these components consistently degrades performance across all metrics, with the most pronounced impact observed when discarding the self-consistency selection mechanism. These results demonstrate that each design choice in SPS plays an important role in achieving the overall effectiveness of the framework.

---

[2]https://github.com/facebookresearch/atlas
[3]https://github.com/AkariAsai/self-rag?tab=readme-ov-file#retriever-setup
[4]https://github.com/princeton-nlp/ALCE

Table 3: Ablation study of SPS on ASQA.

| | ASQA | | |
|---|---|---|---|
| | **em** | **rg** | **mau** |
| SPS$_{\text{Mistral-Instruct-7B}}$ | 27.66 | 32.89 | 66.43 |
| *Staged Retrieval* | | | |
| Always use question $Q$ to retrieve documents | 27.25 (-0.41) | 32.50 (-0.39) | 62.94 (-3.49) |
| *Self-Consistency Selection* | | | |
| Randomly select draft chunk | 26.32 (-1.34) | 25.78 (-7.11) | 58.74 (-7.69) |
| *Muti-pespective sampling* | | | |
| Randomly sampling to form subsets | 26.94 (-0.72) | 31.71 (-1.18) | 63.57 (-2.86) |

Table 4: Effect of Instruction Tuning across datasets.

| | PopQA | TQA | Pub | ARC | ASQA | | |
|---|---|---|---|---|---|---|---|
| | (acc) | (acc) | (acc) | (acc) | (em) | (rg) | (mau) |
| SPS$_{\text{Mistral-7B}}$ | 50.11 | 68.12 | 64.35 | 71.76 | 26.84 | 31.52 | 63.81 |
| SPS$_{\text{Mistral-Instruct-7B}}$ | 51.75 | 70.66 | 67.07 | 73.12 | 27.66 | 32.89 | 66.43 |

**Effect of Staged Retrieval.**     To assess the effectiveness of staged retrieval, we replace it with a static retrieval strategy that always uses the original question $Q$ to retrieve documents, regardless of the current generation context. As shown in Table 3, this modification leads to a consistent drop across all metrics: em decreases by 0.41, ROUGE by 0.39, and mauve by 3.49. These results suggest that dynamically updating the retrieval context during generation helps the model access more relevant evidence, which in turn improves both factual correctness and fluency.

**Effect of Self-Consistency Selection.**     We further evaluate the impact of the self-consistency-based selection strategy by replacing it with a random choice among the generated draft chunks. This change results in substantial performance degradation, particularly in ROUGE (-7.11) and mauve (-7.69), as well as a 1.34-point drop in em. The large decrease in answer quality confirms the importance of self-consistency filtering in identifying the most semantically reliable draft.

**Effect of Multi-Perspective Sampling.**     To examine the role of multi-perspective sampling, we replace it with a random subset sampling strategy. As shown in Table 3, this ablation leads to a moderate decline in performance, with em reduced by 0.72, ROUGE by 1.18, and mauve by 2.86. These results indicate that constructing subsets from diverse perspectives, rather than random sampling, provides more complementary evidence and enhances the robustness of answer generation.

### B.1.1 EFFECT OF INSTRUCTION-TUNING

Although SPS performs well in a training-free setting, we also evaluate it with the instruction-tuned Mistral-Instruct-7B model under the same hyperparameter configuration as in the main results. As shown in Table 4, instruction tuning consistently improves performance across datasets, with gains observed in accuracy on PopQA, TQA, Pub, and ARC, as well as in all three ASQA metrics. This confirms that instruction tuning remains effective for further enhancing our method.

### B.2 EFFECT OF CHUNK SIZE SETTING

As shown in Figure 3, the choice of chunk size affects both performance and system latency. In our framework, the chunk size determines how frequently we perform staged document retrieval and self-consistency-based draft chunk selection. A smaller chunk size results in more frequent selection steps, which improves answer quality by allowing finer-grained control and more timely updates of retrieved context. However, it also increases latency due to the repeated need to encode and

compare multiple draft answers. Conversely, a larger chunk size reduces computational overhead and improves efficiency, but delays both retrieval updates and draft selection, which can hurt performance. Therefore, there exists a trade-off between responsiveness and quality. In our experiments, we find that a chunk size of 50 strikes a good balance, yielding strong performance while keeping latency low.

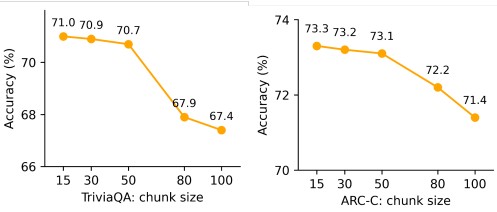 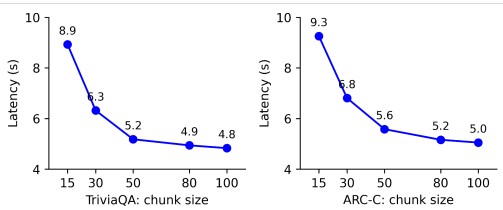

(a) Accuracy under different chunk sizes with $m = 5$, $k = 5$, and top-10 retrieval.

(b) Latency under different chunk sizes with $m = 5$, $k = 5$, and top-10 retrieval.

Figure 3: Performance analysis (a) and latency analysis (b) of SPS$_{\text{Mistral-Instruct-7B}}$ with different chunk size setting on Trivia QA and ARC-Challenge.

### B.3 EFFECT OF DRAFT NUMBER AND DOCUMENT SUBSET SIZE

**Increasing the number of draft chunks per retrieval-generation stage generally leads to improved answer quality.** We conduct experiments under a fixed retrieval setup—retrieving the top-10 documents and setting $k = 5$—while varying the number of drafts. As shown in Figure 4a, we observe a consistent improvement in accuracy as the number of drafts increases. This is because a larger number of drafts allows the model to explore a wider range of document subsets, thereby making more effective use of the retrieved evidence. Moreover, our framework supports launching multiple RAG drafter instances to generate these drafts in parallel, enabling scalability without introducing additional latency.

**Subset Size Does Not Uniformly Improve Performance.** As shown in Figure 4b, increasing the subset size does not necessarily lead to better performance, and its effect largely depends on the characteristics of the dataset. In the case of PopQA, retrieved documents are relatively short, and questions often target specific factual details—e.g., asking about a person's profession or the capital of a country. In such settings, increasing the subset size helps include more relevant information, which benefits answer generation. In contrast, for PubHealth, the retrieved documents tend to be significantly longer, and the task requires verifying whether the evidence supports a given health claim. In this case, a larger subset size may introduce excessive redundancy and irrelevant content, which can overwhelm the model or increase the reasoning complexity. As a result, performance may degrade due to the model being lost in the middle (Liu et al., 2023) of too much information.

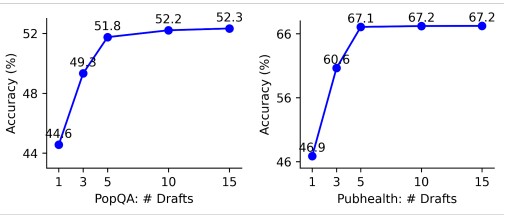 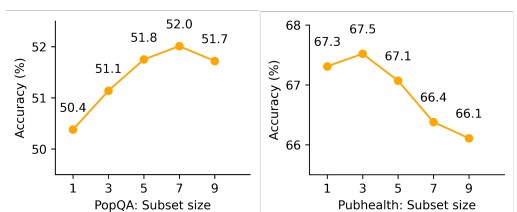

(a) Accuracy with different numbers of drafts, where each subset contains 5 documents. Top-10 documents are retrieved.

(b) Accuracy with different subset sizes, generating 5 drafts per question. Top-10 documents are retrieved.

Figure 4: Performance analysis of SPS$_{\text{Mistral-Instruct-7B}}$ with (a) different numbers of drafts, and (b) different supporting document subset size on PopQA and PubHealth.

### B.4 ANALYSIS OF TOKEN LENGTH AND THROUGHPUT

We calculated the average output token length for different RAG methods and measured the generation speed, where throughput is computed by excluding retrieval time and using a chunk size of 50 tokens. The results are reported in Table 5. Overall, SPS tends to generate longer output sequences, as its staged retrieval mechanism conducts multiple retrieval rounds, thereby incorporating more diverse and informative content into the final response. Speculative RAG also shows relatively longer outputs, since its design often includes additional rationale generation. In contrast, SPS not only produces more comprehensive answers but also consistently achieves significantly higher throughput than all other methods, underscoring its efficiency advantage.

Table 5: Comparison of Avg. Token Length and Throughput across datasets.

| Dataset | Metric | Std.RAG (Mistral-7B) | Self-RAG (Mistral-7B) | Spec.RAG (8×7B, 7B) | SPS (Mistral-7B) |
|---|---|---|---|---|---|
| PopQA | Avg. Token Length | 85.8 | 92.5 | 113.1 | 106.7 |
|  | Throughput (tokens/s) | 16.0 | 10.9 | 9.1 | 20.5 |
| TriviaQA | Avg. Token Length | 82.1 | 91.3 | 108.4 | 105.0 |
|  | Throughput (tokens/s) | 15.9 | 10.4 | 9.2 | 20.3 |
| PubHealth | Avg. Token Length | 101.5 | 103.9 | 132.3 | 129.6 |
|  | Throughput (tokens/s) | 18.1 | 12.7 | 11.8 | 23.9 |
| ARC-C | Avg. Token Length | 90.6 | 96.2 | 117.1 | 114.6 |
|  | Throughput (tokens/s) | 17.8 | 11.6 | 9.0 | 20.5 |
| ASQA | Avg. Token Length | 123.7 | 125.6 | 175.2 | 163.5 |
|  | Throughput (tokens/s) | 16.4 | 10.2 | 9.0 | 21.5 |

### B.5 EFFECT OF HARDWARE ON LATENCY

In the main results, we conducted experiments on a server equipped with 4 RTX 6000 Ada GPUs. The part of high latency of Speculative RAG primarily stems from its reliance on larger models and the associated multi-GPU inference, where the 6000 Ada exhibits considerable overhead in inter-GPU communication. To further examine the impact of hardware, we repeated the experiments on 4 Nvidia A100-SXM4-40GB GPUs, which leverage NVLink to provide substantially faster multi-GPU communication. As shown in Figure 5, Speculative RAG's latency decreases markedly under this setup. Nevertheless, across all datasets, our SPS method consistently achieves significantly lower latency than Speculative RAG, highlighting its robustness and efficiency regardless of the underlying hardware.

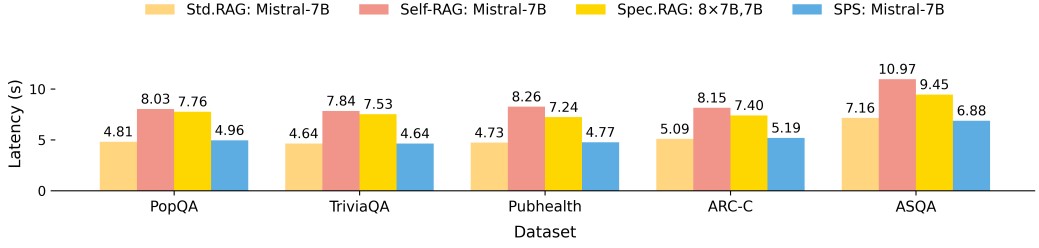

Figure 5: Latency analysis using 4 Nvidia A100-SXM4- 40GB GPUs.

## C  A SIMPLE EXAMPLE OF STAGED RETRIEVAL

A single question posed to large language models may span multiple topics. In traditional RAG frameworks, document retrieval is performed only once before answer generation begins. As a result, the retrieved documents may fail to cover all relevant subtopics or domain-specific information required to generate a comprehensive response. In contrast, periodic retrieval enables the system to retrieve new documents at intermediate stages of generation, after each topic-specific segment is produced. This allows the retrieval process to remain closely aligned with the current focus of the response, ensuring that each part of the answer is supported by highly relevant information. Consequently, periodic retrieval significantly improves the factual coverage and coherence of the generated answers.

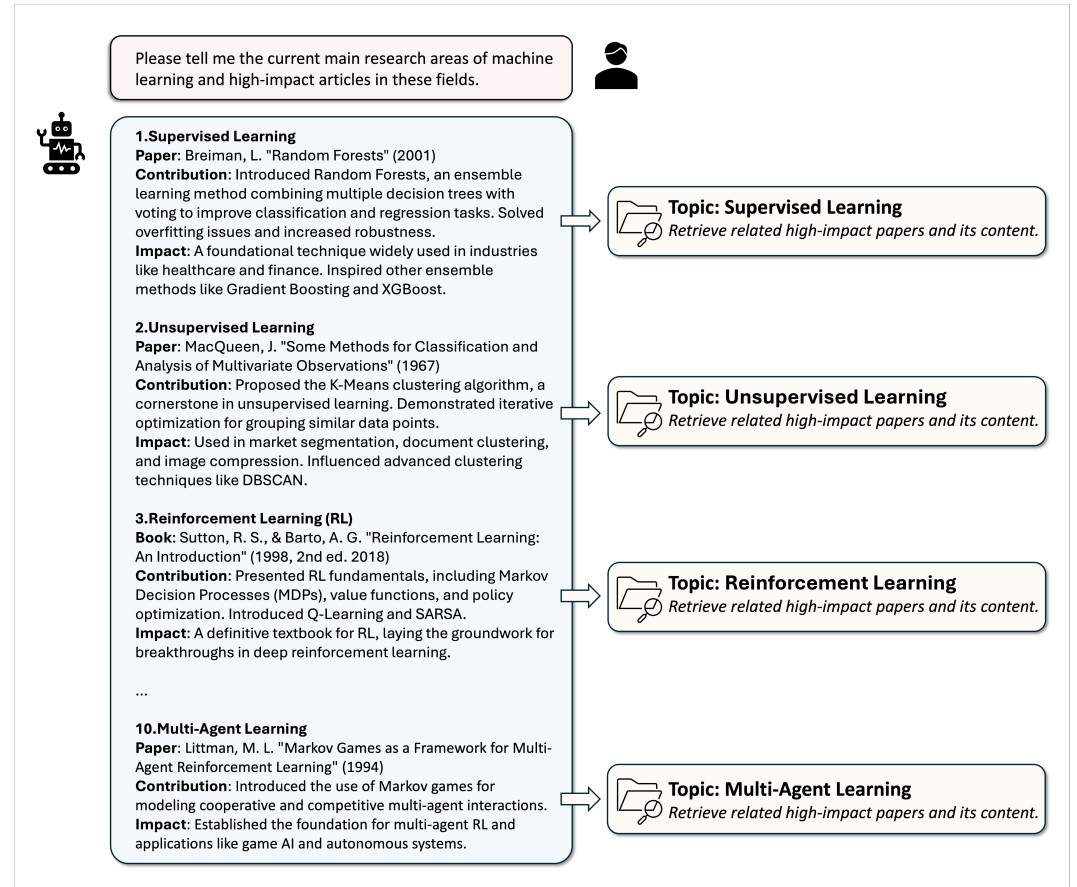

Figure 6: A simple example of staged retrieval.

# D CASE STUDY

```
=============================== Prompt ================================
##Instruction:
Write a clear, informative, and balanced answer to the following
question, based on the documents below. If the question is ambiguous or
 has multiple interpretations, explain the possible answers based on
the context provided.

##Question:
Who's the highest paid nba player 2017?

##Retrieval Documents:

1. Highest-paid NBA players by season

has earned on a 1 year contract, Jordan also holds the record for the
second largest 1 year contract at $30,140,000 in the 1996-97 season.
Kobe Bryant become just the second player to reach this milestone when
the 2013-14 season began. LeBron James became the third in the 2016-17
season. Stephen Curry became the first player to eclipse $40-Million
per year when he signed a record 5-year contract worth $201-Million in
2017, starting with $34,682,550 in the 2017-18 season and ending with
the largest earnings in the 2021-22 season with a record payout of $45
,780,966.

2. NBA salary cap
signed the contract once the NBA's free agency moratorium ended on July
 6, 2017. Shortly thereafter, James Harden agreed on a DVPE with the
Rockets. At the time of signing, his current contract had two years
remaining with total pay of $59 million; the extension will add another
 $170 million over four seasons, ending in 2022-23. The next supermax
signing was that of John Wall, who agreed later in July to a four-year,
 $170 million extension that will start in 2019-20. In late September,
Russell Westbrook became the fourth and final supermax signing of the
2017 offseason, signing a five-year,

...

10. Chris Paul
a 112-102 win. He also had 10 assists and seven rebounds. In Game 5 of
the Western Conference Finals, Paul injured his hamstring late in the
game as the Rockets went up 3-2 in the series against the Warriors.
With Paul out for Games 6 and 7, the Rockets were eliminated from the
playoffs with back-to-back losses.On July 8, 2018, Paul signed a four-
year, $160 million maximum contract extension with the Rockets. Paul
received a two-game suspension early in the 2018-19 season for his
involvement in an on-court fight against the Los Angeles Lakers on
October 20. Additionally, Paul

========================== Cluster results ===========================
K-means clustering...
Group1:[1,3]
Group2:[2,4]
Group3:[5,7]
Group4:[6,10]
Group5:[8,9]
========================= generate subsets ==========================
Sample from each group...
Subset1:[1,2,5,6,8]
Subset2:[1,4,7,10,9]
Subset3:[3,4,7,6,8]
Subset4:[1,2,7,10,9]
Subset5:[3,4,5,6,9]

======================= generate draft chunks =======================
1.Based on the information provided in the documents, the highest paid
NBA player in 2017 could be LeBron James,Stephen Curry, or James Harden.
 LeBron became only the third player ever to surpass the $30 million
```

```
single season threshold during the 2016-17 campaign, putting him at the
 top of that year's payroll list...

2.Based on the information provided in the documents, the highest-paid
NBA player in 2017 could be LeBron James, Stephen Curry, or James
Harden. LeBron became only the third player ever to surpass the $30-
million single-season threshold during the 2016-17 campaign, putting
him at the top of that year's payroll list...

3.Based on the documents, the highest-paid NBA player in 2017, by
actual 2017-18 salary, was Stephen Curry, whose record-setting five-
year $201-million super-max began paying him $34,682,550 that October.
He was the first player ever to eclipse $40-million per year, setting a
 new standard for league earnings...

4.Based on the information provided in the documents, the highest-paid
NBA player in 2017 could be LeBron James, Stephen Curry, or James
Harden. LeBron became only the third player ever to surpass the $30-
million single-season threshold during the 2016-17 campaign, putting
him at the top of that year's payroll list...

5.Based on the available documents, the highest-paid NBA player in 2017,
 when counting the 2016-17 season salary, was LeBron James, who became
the third man in history to top $30-million for a single season and
finished that campaign asnthe league-s biggest earner beforenfree-
agency began in July that year...

--
Based on sematic similarity select chunk 1 and continue generating...

...

====================== generate final answer ========================
##Response:
Based on the information provided in the documents, the highest-paid
NBA player in 2017 could be LeBron James, Stephen Curry, or James
Harden. LeBron became only the third player ever to surpass the $30-
million single-season threshold during the 2016-17 campaign, putting
him at the top of the pay scale as that season ended. Shortly after the
 2017 free-agency moratorium lifted, Stephen Curry signed a record five-
year, $201 million contract whose first-year payout-$34,682,550 in
2017-18, which made him the first player to eclipse $40 million per
season. Days later, James Harden agreed to a four-year, $170 million
extension with Houston, part of a package totaling roughly $228 million
 in guaranteed money, the largest deal inked that calendar year.

The documents do not provide a single, definitive salary table for 2017,
 but they do note a soft salary cap of $99 million and a luxury-tax
line of $119 million for the 2017-18 season. Therefore, depending on
whether one emphasizes salary already earned, the first paycheck of the
 new season, or the biggest contract signed, any of these three stars
can reasonably be called the highest-paid NBA player of 2017.
```

Table 6: Full list of instructions used during zero-shot evaluations.

| Dataset | Instruction |
|---|---|
| TriviaQA | Use the evidence documents to answer the following question. If the documents do not provide enough information, try to answer with your own knowledge and clearly indicate that this is not directly supported by the documents. |
| PopQA | Use the evidence documents to answer the following question. If the documents do not provide enough information, try to answer with your own knowledge and clearly indicate that this is not directly supported by the documents. |
| ARC-Challenge | Use the evidence documents to answer the following choice question. Given four answer candidates, A, B, C and D, choose the best answer choice. |
| PubHealth | Given the following claim and a set of evidence documents, determine whether the claim is: - SUPPORTS (evidence clearly supports the claim) - REFUTES (evidence clearly contradicts the claim) Please reason carefully based only on the provided evidence. Do not use any external knowledge. |
| ASQA | Write a clear, informative, and balanced answer to the following question, based on the documents below. If the question is ambiguous or has multiple interpretations, explain the possible answers based on the context provided. |

## E  LLM USAGE DISCLOSURE

LLMs are employed only to aid writing clarity and polish. Importantly, all core scientific contributions, including problem formulation, model design, theoretical analysis, and experiments, are entirely conceived and executed by the authors. The authors take full responsibility for all technical content, claims, and conclusions presented in this work.

