# OpenReview forum: "Training-Free Speedup for Retrieval-Augmented Generation with Staged Parallel Speculation"
_ICLR.cc/2026/Conference — Submitted to ICLR 2026_

### Official Review · Reviewer_osbW · 2025-10-29

**Soundness:** 3
**Presentation:** 3
**Contribution:** 3
**Rating:** 6
**Confidence:** 4

**Summary:**

This paper proposes parallel generation to accelerate the generation process in RAG pipeline.

**Strengths:**

* Novel idea of retrieving the documents on-the-fly.
* Contains latency evaluation
* Higher accuracy in evaluation

**Weaknesses:**

* Evaluation focuses on QA dataset.
* Missing motivating examples and some technical details.
* No throughput evaluation.

**Questions:**

* What is the need to change the retrieved document on the fly instead of simply retrieve multiple documents related to the query? Is there a motivating example? I understand that context may shift during generation process, but the shifted context is still answering the original query and thus still have high semantic similarity to the original query.
* How do you perform the generation M.generate(C, sj)? Are you customizing the chat template to incorporate sj, or just directly append sj to the original answer? This is important because different ways of inserting sj greatly effects the efficiency of underlying inference engine (e.g. if you insert sj at the end of the generation, the generation quality may drop (as you are forcing the LLM to continue completing sj) but it is very fast because it can leverage the prefix cache).
* What about question answering throughput? I am mentioning this because it seems to me that your approach is trading throughput for latency. If that's the case, make it explicit, if that's not the case, would love to hear more about the rationale.

---

> ### Author Response · Authors · 2025-11-21
>
> > **W1:** Evaluation focuses on QA dataset.
>
> **R1:** Thank you for bringing up this point. Our work primarily focuses on QA tasks because RAG is fundamentally designed to address questions that require external knowledge sources, and most established benchmarks in this area are also QA-oriented. Likewise, **both** of our main baselines, Self-RAG [2] and Speculative RAG [1], evaluate their methods only on QA tasks. We therefore follow the same setup to ensure a fair and consistent comparison. We appreciate the suggestion to expand the evaluation beyond QA datasets, and we leave this exploration for future work.
>
>
>
>
> > **W2,W3:** Missing motivating examples and some technical details. No throughput evaluation.
>
> **R2:** We would like to kindly point out that we have already provided the relevant information in Appendix. Specifically, we kindly refer you to **Appendix A** for the technical details, **Appendix B.4** for the throughput experiments, **Appendix C** for motivating examples. The throughput experiments report the average token length under each method and compute the corresponding throughput (tokens/s). The results show that our SPS method **consistently outperforms** the other baselines in terms of throughput.

---

> ### Author Response · Authors · 2025-11-21
>
> > **Q1:** What is the need to change the retrieved document on the fly instead of simply retrieve multiple documents related to the query? Is there a motivating example? I understand that context may shift during generation process, but the shifted context is still answering the original query and thus still have high semantic similarity to the original query.
>
> **R3:** Thank you for raising this question. We provide a simple illustrative example in Appendix C. Specifically, with one-shot retrieval, the most likely matched keyword may be “machine learning” in that example. In such cases, the retrieved documents can **concentrate** on some most related subfields—for example, LLMs—resulting in incomplete coverage of the relevant topics needed for the final answer.
>
> In contrast, **staged retrieval** incorporates the partially **generated answer** back into the retrieval process. When the model begins discussing a particular paper or subfield, the semantic similarity naturally steers the retrieval results toward that **specific domain** or even toward **details** of that specific paper. This adaptive refinement allows the model to access more targeted and contextually relevant information, ultimately producing a more comprehensive and accurate answer.
>
>
> > **Q2:** How do you perform the generation M.generate(C, sj)? Are you customizing the chat template to incorporate sj, or just directly append sj to the original answer? This is important because different ways of inserting sj greatly effects the efficiency of underlying inference engine (e.g. if you insert sj at the end of the generation, the generation quality may drop (as you are forcing the LLM to continue completing sj) but it is very fast because it can leverage the prefix cache).
>
>
> **R4:** Thank you for raising this concern. We would like to note that we adopt the exact **same chat template** as Speculative RAG [1] to incorporate sj, namely in the order of *Question → retrieved documents → answer sequence* (as shown in Appendix C.1). This template structure forces the LLM to read sj **before** continuing answer generation, i.e., the generation M.generate(C, sj) starts to continue sampling the next answer chunk **directly** at the end of the previous generated chunk, rather than completing sj. Therefore, in each generation round, we **do not use** the prefix cache for efficiency.
>
>
> > **Q3:** What about question answering throughput? I am mentioning this because it seems to me that your approach is trading throughput for latency. If that's the case, make it explicit, if that's not the case, would love to hear more about the rationale.
>
>
> **R5:** Thank you for raising this concern regarding throughput. We believe the throughput refers to the **number of tokens generated in 1 second**. If this interpretation differs from yours, we would be happy to clarify further in the next discussion.
>
> We kindly refer you to the throughput experiments in **Appendix B.4**, whose results demonstrate that our SPS method consistently achieves **higher** token-generation rates than the other baselines.
>
> Regarding the comment that our method “trades throughput for latency,” we would like to note that what SPS actually trades is **computation overhead** for lower latency and higher accuracy. Latency reduction primarily comes from the parallelized design of our retrieval and generation stages. As shown in the breakdown analysis provided in our response **R4** to `Reviewer FtgG`, parallelization minimizes the idle waiting between the retrieval subsystem and the LLM inference subsystem, effectively reducing end-to-end latency.
>
> Accuracy improvement is mainly due to generating multiple **parallel answer paths** and selecting the best one through **self-consistency**, as we show on ablation study in Appendix B.1. The additional computation overhead introduced by this step comes from sampling these parallel paths. When generating 5 parallel paths, the actual computational overhead will be **less than** 5 times that of Standard RAG, because each parallel path in our method operates with a significantly shorter context.
>
> **References:**
>
> [1] Zilong Wang, et al. "Speculative rag: Enhancing retrieval augmented generation through drafting." International Conference on Learning Representations, 2024.
>
> [2] Akari Asai, et al. Self-RAG: Learning to retrieve, generate, and critique through self-reflection. In The Twelfth International Conference on Learning Representations, 2024.

---

> ### Author Response · Authors · 2025-11-27
>
> Dear Reviewer osbW,
>
> We would like to express our sincere gratitude for your thoughtful review and valuable feedback. We believe we have carefully addressed all the questions and concerns raised in your comments in our responses, and we hope our clarifications help resolve the issues you pointed out.
>
> If you have any further questions or would like additional clarification on any point, we would be very glad to respond promptly. Thank you again for your time and consideration.
>
> Best regards,
>
> All authors

---

### Official Review · Reviewer_brDz · 2025-11-01

**Soundness:** 2
**Presentation:** 2
**Contribution:** 2
**Rating:** 4
**Confidence:** 3

**Summary:**

This paper presents Staged Parallel Speculation (SPS), a framework for RAG that achieves latency reduction by running inference and retrieval in parallel without additional training. SPS runs retrieval periodically after a fixed number of generated tokens and clusters the retrieved documents by content similarity, sampling one from each cluster to maximize diversity while reducing context length. It then generates answer candidates in parallel and selects the final answer leveraging self-consistency. It reduces RAG latency by 57% while reaching 96% of the accuracy of finetuning-based methods.

**Strengths:**

- Latency reduction of RAG applications without additional fine-tuning.

**Weaknesses:**

- The accuracy is not as good as fine-tuning-based methods (Table 1)
- The efficiency comparison against Speculative RAG is not fair since the model size is different.
- The efficiency evaluation is limited to the case of batch size 1. It is unclear whether the proposed method offers a throughput benefit in a batching scenario, which is common in data center serving scenarios.

**Questions:**

- What is the overhead of the proposed method, including additional computation and memory requirements due to additional LLM and retrieval calls?
- The SPS algorithm seems overly complex, and it is unclear how much each component contributes to the overall performance. Can you provide an ablation study, such as disabling part(s) of the algorithms (clustering, self-consistency selection, staged retrieval, etc) and other hyperparameters, such as retrieval amount?
- How good is the SPS's performance compared to other efficiency optimization methods for RAG, including PipeRAG and many other works [1-5]?

### References

- [1] https://arxiv.org/abs/2404.12457
- [2] https://arxiv.org/abs/2405.16444
- [3] https://arxiv.org/abs/2502.20969
- [4] https://arxiv.org/abs/2503.14649
- [5] https://arxiv.org/abs/2505.07833

---

> ### Author Response · Authors · 2025-11-21
>
> We really appreciate your positive and instructive feekback and will provide detailed responses below.
>
> > **W1:** The accuracy is not as good as fine-tuning-based methods (Table 1)
>
> **R1:** Although our method does not exceed the accuracy of the strongest fine-tuning–based approaches, it reaches over 90% of their performance and even surpasses certain fine-tuned baselines (e.g., CRAG [1]). Importantly, SPS is entirely **training-free**, which gives it a key advantage in terms of applicability and efficiency. Fine-tuning–based methods typically perform well on the specific datasets they are trained on, but their performance often **degrades significantly on unseen datasets**[2]. In contrast, our training-free approach maintains strong accuracy across all evaluated datasets, demonstrating its robustness and wide applicability. Moreover, eliminating the need for training also **saves substantial computational cost** and enables true **plug-and-play** usage. This advantage has also been explicitly acknowledged in the strengths highlighted by `Reviewer DhGh` and `Reviewer FtgG`.
>
>
> > **W2:** The efficiency comparison against Speculative RAG is not fair since the model size is different.
>
> **R2:** Thank you for raising the concern regarding fairness in comparison. We would like to note that our method and Spec.RAG use **the same generation model**, *Mistral-7B*. Spec.RAG additionally requires a stronger verify model as an **essential component of its design** [3], and we retain this setting to ensure methodological fidelity. Therefore, our comparison follows the original design choices of Spec.RAG and remains fair and rigorous.
>
> > **W3:** The efficiency evaluation is limited to the case of batch size 1. It is unclear whether the proposed method offers a throughput benefit in a batching scenario, which is common in data center serving scenarios
>
> **R3:** Thank you for bringing up this point. SPS focuses on algorithm-level acceleration. The computationally intensive components of SPS all **naturally support batching and parallel computation** in modern inference frameworks:
>
> 1.Dense retriever encoders (e.g., BERT-style) are fully batch-parallel.
>
> 2.The distance computations in k-means clustering are vectorized and efficiently parallelizable.
>
> 3.LLM inference and sampling similarly benefit from batching in standard inference engines.
>
> Therefore, SPS can remains effective under batching: the **parallelizable nature** of its **core components** allows SPS to provide speedup even when processing multiple queries simultaneously.

---

> ### Author Response · Authors · 2025-11-21
>
> > **Q1:** What is the overhead of the proposed method, including additional computation and memory requirements due to additional LLM and retrieval calls?
>
> **R4:** We appreciate your concern regarding additional cost. We acknowledge that our method introduces more computational overhead compared to standard RAG. This is an **inherent trade-off** of our design: we intentionally allocate more computation to achieve **substantially lower latency** (via parallelization) and higher answer accuracy.
>
> Although using m=5 subsets gives a theoretical upper bound of **up to** 5× the compute of standard RAG, the practical overhead is **notably smaller** because each parallel path processes far fewer retrieved documents, leading to much shorter prompts and reduced per-path workload.
>
> Regarding memory usage, under **batch inference**, only a single model instance is required; the additional memory primarily comes from caching and storing hidden states for parallel paths. While the worst-case upper bound could approach 5× when context lengths are extremely long, in practice the context is much shorter, and the observed memory usage is only about 1–2× that of the standard RAG baseline.
>
> As for the **retrieve calls**, we evaluated the average **retrieval frequency** per question on the PopQA dataset with *Mistral-7B* as generator and *Contriver-MS MARCO* as retriever. The results are as follows:
>
> ||Standard RAG|Self-RAG|Spec.RAG|SPS|
> |---|---|---|---|---|
> |Avg retrieve times|1|1.4|1|1.7|
>
> Spec.RAG and Standard RAG each perform only a single retrieval before generation, while both SelfRAG and our method may trigger additional retrieval during generation. Although SPS has the highest retrieval count, it is still acceptable, indicating that our approach **does not incur** excessive retrieval overhead. Moreover, since retrieval models are **significantly smaller** than the LLM, retrieval cost is not the dominant source of computational cost in RAG systems.
>
>
> > **Q2:** The SPS algorithm seems overly complex, and it is unclear how much each component contributes to the overall performance. Can you provide an ablation study, such as disabling part(s) of the algorithms (clustering, self-consistency selection, staged retrieval, etc) and other hyperparameters, such as retrieval amount?
>
> **R5:** Thank you for expressing your concerns regarding the ablation studies and the selection of other hyperparameters. First, we would like to kindly point out that we have already provided ablation studies in the initial submmision. Please refer to **Appendix B.1**, which analyzes the effects of staged retrieval, self-consistency selection, and multi-perspective sampling. As reported in Table 3, **self-consistency selection contributes the most to performance improvements**, while staged retrieval and multi-perspective sampling offer moderate but still meaningful gains.
>
> In addition, we also examine the impact of **hyperparameter choices**: such as the chunk size, subset size, number of draft answers. More details are in **Appendices B.2 and B.3**.
>
>
> > **Q3:** How good is the SPS's performance compared to other efficiency optimization methods for RAG, including PipeRAG and many other works [1-5]?
>
> **R6:** We thank you for suggesting additional valuable baselines.
>
> Regarding PipeRAG, it is **not an LLM-based RAG** method, which is why it was not included in our initial comparisons. We now implement a PipeRAG-style baseline, and we kindly refer you to our **R1** to `Reviewer FtgG` for detailed experimental results and comparison.
>
> Regarding the other works you mentioned, they are indeed valuable contributions. However, their optimization goals **differ from ours**: [1,2] accelerate RAG primarily through **KV-cache–level** optimizations. [3] targets RAG methods under the **specific setting** of **query rewriting**. [4,5] focus on **large-scale RAG serving**, improving throughput via **hardware-aware resource scheduling**.
>
> In contrast, our method focuses on **algorithmic-level** optimization for RAG. Therefore, the direction of optimization in our approach is **orthogonal** to these works, making a direct comparison with SPS not entirely fair or particularly meaningful. In fact, SPS could be combined with several of these methods to achieve even greater speedups.
>
> That said, we sincerely appreciate you for pointing out these valuable works, and we will include them in the references of the final version.
>
> **References:**
>
> [1] Shi-Qi Yan, et al. "Corrective retrieval augmented generation." Arxiv preprint, 2024.
>
> [2] Zhenyi Wang, et al. "A Comprehensive Survey of Forgetting in Deep Learning Beyond Continual Learning." Transactions on Pattern Analysis and Machine Intelligence, 2023.
>
> [3] Zilong Wang, et al. "Speculative rag: Enhancing retrieval augmented generation through drafting." International Conference on Learning Representations, 2024.

---

> > ### Comment · Reviewer_brDz · 2025-11-27
> >
> > Thanks for your response. I have some follow-up questions.
> >
> > ### W2
> >
> > For the efficiency results shown in Figure 2 and Table 2, can you provide data for Speculative RAG with Mistral-7B as the verifier (as used in Table 1)? I think that gives a better idea about the efficiency advantage of your method.
> >
> > ### Q1
> >
> > Is it possible to quantify the compute and memory overhead due to the additional LLM calls? If the LLM is the dominant source of computational cost, it is more important to quantify the overhead in the LLM part.

---

> > > ### Author Response · Authors · 2025-12-02
> > >
> > > Thank you for your follow-up questions. We provide detailed responses below.
> > >
> > > > **W1:** For the efficiency results shown in Figure 2 and Table 2, can you provide data for Speculative RAG with Mistral-7B as the verifier (as used in Table 1)? I think that gives a better idea about the efficiency advantage of your method.
> > >
> > > **R1:** We present the relevant efficiency results here:
> > > Latency: time(seconds)
> > > |Method|PopQA|TQA|Pub|ARC|ASQA|
> > > |---|---:|---:|---:|---:|---:|
> > > |Spec.RAG(7B,7B)|6.31|6.49|6.73|6.82|8.84|
> > > |SPS(7B)|5.21|5.18|5.44|5.58|7.61|
> > >
> > > Average latency and performance across 4 datasets
> > > |RAG Method|Training|Latency(s)|Acc(%)|
> > > |---|---|---|---|
> > > |Spec.RAG(7B,7B)|No|8.80|44.42|
> > > |SPS(7B)|No|5.35|63.59|
> > >
> > > As shown, even when Speculative RAG uses a 7B-size model as the verifier, **SPS achieves lower latency and higher accuracy** under the same LLM.
> > > The lower latency mainly comes from the fact that SPS does not require generating additional rationales, and our self-consistency–based selection remains more efficient than verifying drafts with a 7B model. The higher accuracy is largely attributed to the effectiveness of our staged retrieval and self-consistency selection modules.
> > >
> > > > **W2:** Is it possible to quantify the compute and memory overhead due to the additional LLM calls? If the LLM is the dominant source of computational cost, it is more important to quantify the overhead in the LLM part.
> > >
> > > **R2:** Thank you for raising this concern regarding the quantification of computational overhead. We have added additional experiments, including actual memory usage under batch inference and FLOPs analysis.
> > >
> > > **1. Memory Overhead**
> > > The primary contributor to memory consumption is the model allocation itself. In our setup, the allocated model sizes are: **Mistral-7B (13.50 GB)** and **Mixtral-8×7B (27.03 GB)**.
> > > We further measured the actual runtime memory usage (Speculative RAG and SPS both use batch inference).
> > >
> > > |RAG Method|Memory(GB)|
> > > |---|---|
> > > |Standard RAG(Mistral-7B)|13.81|
> > > |Self-RAG(Mistral-7B)|14.24|
> > > |Spec.RAG(Mixtral-8×7B,Mistral-7B)|41.85|
> > > |SPS(Mistral-7B)|14.37|
> > >
> > > From these results, we observe that:
> > > - Compared to Standard RAG, the memory overhead introduced by additional LLM calls in SPS is 14.37 − 13.81 = 0.56 GB, which is negligible relative to the overall LLM memory footprint.
> > > - Our memory usage is comparable to Self-RAG, and far lower than standard Speculative RAG.
> > >
> > > Overall, **model loading dominates memory consumption**, not the extra LLM calls, as batch inference effectively amortizes this overhead.
> > >
> > > ---
> > >
> > > **2. Computational Overhead (FLOPs)**
> > > We computed the actual input/output tokens for different RAG methods on PopQA, and derived FLOPs using the Mistral family configuration (hidden size = 4096, intermediate size = 14336). The results are as follows:
> > >
> > >
> > > |RAG Method|Input tokens|Output tokens|FLOPs $(× 10^{13})$|
> > > |---|---|---|---|
> > > |Standard RAG(Mistral-7B)|1459|86|1.94|
> > > |Pipe-Style RAG(Mistral-7B)|2612|90|3.56|
> > > |Self-RAG(Mistral-7B)|2014|93|2.71|
> > > |Spec.RAG(Mixtral-8×7B,Mistral-7B)|3725(draft)+4290(verify)|113|14.78|
> > > |Spec.RAG(Mistral-7B,Mistral-7B)|3725(draft)+4281(verify)|111|13.52|
> > > |SPS(Mistral-7B)|7823|107|12.61|
> > >
> > > These results show that although SPS incurs higher FLOPs than baselines like Standard RAG, it still remains **lower than Speculative RAG** (for both small and large verifier models). This highlights the key trade-off of our method: SPS incurs moderately higher computation to deliver lower latency and higher accuracy, while still maintaining a lower overall cost than Speculative RAG. Its efficiency stems from avoiding the generation of additional rationales and replacing verifier-based checking with a lightweight self-consistency selection mechanism.

---

### Official Review · Reviewer_FtgG · 2025-11-03

**Soundness:** 2
**Presentation:** 3
**Contribution:** 2
**Rating:** 2
**Confidence:** 2

**Summary:**

The paper addresses the latency issue in RAG systems. In a standard RAG setup, you have a LLM that, given a query, first retrieves documents from a knowledge base then uses those documents as context to generate an answer. Retrieving and then waiting for generation introduces pauses and idle time. The authors propose a new training-free framework SPS which is designed to significantly speed up RAG inference while preserving accuracy. Different from standard RAG, the retrieval system runs in parallel with the generation system so that the generation process does not wait for retrieval as a blocking step. On multiple QA benchmarks, SPS shows lower latency compared to other training-free RAG baselines, while achieving similar accuracy with fine-tuned RAG methods.

**Strengths:**

* Unlike other RAG acceleration methods such as SpecRAG, SPS requires no model retraining or fine-tuning. This makes it immediately deployable on existing pipelines and models, which is a major practical advantage.
* SPS approaches the problem with overlapped retrieval and generation, eliminating the “wait gap” between the two.
* Experimental results show large and consistent latency reduction.

**Weaknesses:**

* While the paper frames SPS as a training-free speedup for RAG, the core idea is conceptually close to prior papers such as PipelineRAG and SpecRAG. I recommend adding more detailed evaluation and discussion with identical retrievers, LLMs and datasets and report wall-clock latency as well as accuracy.
* The paper does not report detailed ablation studies. It would be beneficial to show the performance without clustering; varying candidates and stage length; swap the self-consistency selector with others.
* Results primarily focus on mid-sized open LLMs. It remains unclear whether SPS scales to larger models or interacts differently with stronger retrievers.
* The paper only reports aggregate latency but lacks a decomposed timeline showing where savings arise. Without breakdowns, it is difficult to assess the mechanism’s effectiveness.

**Questions:**

N/A

---

> ### Author Response · Authors · 2025-11-21
>
> We really appreciate your positive and instructive feekback and will provide detailed responses below.
>
> > **W1:** While the paper frames SPS as a training-free speedup for RAG, the core idea is conceptually close to prior papers such as PipelineRAG and SpecRAG. I recommend adding more detailed evaluation and discussion with identical retrievers, LLMs and datasets and report wall-clock latency as well as accuracy.
>
> **R1:** Thank you for raising this insightful point. We clarify our comparisons with both PipeRAG and Spec.RAG and highlight how SPS differs conceptually and practically from these methods. This disscussion will be add to the revised paper shortly. (We assume “PipelineRAG” refers to PipeRAG [1]; if this is a misunderstanding on our part, we would be happy to clarify it further in the next discussion round.)
>
> - **Comparison with PipeRAG**
>     - **Different architectural assumptions.** As discussed in Sec. 2.1, PipeRAG is **built on RETRO** [2], a jointly trained encoder–decoder architecture, and its gains come from parallelizing retrieval and generation **within that joint model**. In contrast, SPS is designed specifically for decode-only LLMs, where such joint training is not available. Thus, PipeRAG’s original mechanism cannot be directly applied to LLM-based RAG.
>     - **Key Innovations Beyond PipeRAG.** As we highlighted in Sec. 3.3, SPS **extends** the fixed-chunk retrieval scheme in PipeRAG into an existing chunk sequence retrieval procedure $(C_1,C_2,… )$, that better aligns with the semantics of modern dense retrievers and supports LLM-based generation. Furthermore, SPS introduces an innovative use of **multi-perspective sampling** together with **self-consistency** to select the best chunk from the parallel candidates, thereby improving the overall chunk-generation quality.
>
>     - **Fair and aligned evaluation.** To ensure a fair comparison with PipeRAG, we additionally implement a PipeRAG-style LLM-based baseline using the same retriever as SPS, Self-RAG, and SpeculativeRAG, the same LLM generator (*Mistral-7B*), and the same chunk size (50). The results are shown in the table below:
>
>         Performance: accuracy (%)
>         |Method|PopQA|TQA|Pub|ARC|
>         |--|--|--|--|--|
>         |Standard RAG|32.59|53.50|35.26|43.51|
>         |PipeRAG-style RAG|32.47|54.16|34.19|41.28|
>         |SPS|50.11|68.12|64.35|71.76|
>
>         Time: latency (s)
>         |Method|PopQA|TQA|Pub|ARC|
>         |--|---|--|--|--|
>         |Standard RAG| 5.36|5.15|5.62|5.09|
>         |PipeRAG-style RAG|5.59|5.34|5.78|5.73|
>         |SPS|5.21|5.18|5.44|5.58|
>
>         Results show that the performance of PipeRAG-style RAG is roughly **on par** with standard RAG, highlighting that SPS’s improvements arise **not from generic parallelism alone**, but from our chunk-sequence sampling + self-consistency mechanism, designed specifically for LLM-based RAG.
>
> - **Comparison with Spec.RAG**
>
>     - **Key Innovations Beyond Spec.RAG.** SPS draws inspiration from Spec.RAG’s multi-perspective sampling, while introducing a **self-consistency mechanism** for draft selection. Unlike Spec.RAG, SPS requires **no additional training** (e.g., no rationale generation) and requires **no larger verifier LLM** for filtering, making it immediately deployable on existing pipelines and models.
>
>     - **Fair and aligned evaluation.** We would like to note that we **follow the exact configuration of the original Spec.RAG evaluation** [3]. Specifically, we adopt the *Mistral* family of LLMs, the *Contriever-MSMARCO* retriever, and multiple datasets (*PopQA, PubMedQA, TQA, and ARC*) **fully consistent with** the evaluation setting adopted by Spec.RAG [3] to ensure a fair comparison, and we report both wall-clock latency and accuracy (please refer to Table 1 and Figure 2 in our paper).
>
> In summary, SPS **combines the strengths** of PipeRAG and Spec.RAG while **introducing adaptations** that make it more suitable for LLM-based RAG.
>
> > **W2:** The paper does not report detailed ablation studies. It would be beneficial to show the performance without clustering; varying candidates and stage length; swap the self-consistency selector with others.
>
> **R2:** Thank you for expressing your concerns regarding the ablation studies and the selection of other hyperparameters. First, we would like to kindly point out that we have provided ablation studies in our initial submmission. Please refer to **Appendix B.1**, which analyzes the effects of staged retrieval, self-consistency selection, and multi-perspective sampling. As reported in Table 3, **self-consistency selection contributes the most to performance improvements**, while staged retrieval and multi-perspective sampling offer moderate but still meaningful gains.
>
> In addition, we also examine the impact of hyperparameter choices: the **stage length** (i.e., chunk size) and the **number of candidates** (i.e., draft answers) in **Appendices B.2 and B.3**. We hope these results address your concerns.

---

> ### Author Response · Authors · 2025-11-21
>
> > **W3:** Results primarily focus on mid-sized open LLMs. It remains unclear whether SPS scales to larger models or interacts differently with stronger retrievers.
>
> **R3:** We appreciate your concern regarding the scalability and stronger retrievers. To address this, we have added experiments using a larger LLM (*Mixtral-8×7B*), keeping all other settings identical except for the model. The results are shown below:
>
> |Method|PopQA|TQA|Pub|ARC|
> |---|---|---|---|---|
> |SPS(Mistral-7B)|50.11|68.12|64.35|71.76|
> |SPS(Mixtral-8×7B)|54.20|71.03|70.97|76.93|
>
> In addition, we conducted further experiments on PopQA and TQA using a stronger retriever *Qwen3-Embedding-0.6B*, while keeping the LLM fixed to *Mistral-7B*. The results are as follows:
>
> |Retriever|PopQA|TQA|
> |---|---|---|
> |Contriever-MS MARCO|50.11|68.12|
> |Qwen3-Embedding-0.6B|50.64|68.80|
>
> From the results, we observe **consistent accuracy improvements** when using a larger LLM or a stronger retriever. The gains from scaling up the LLM are more pronounced, while the improvements from using a stronger retriever are comparatively smaller. These findings demonstrate the **broad applicability** and **robustness** of our method.
>
> > **W4:** The paper only reports aggregate latency but lacks a decomposed timeline showing where savings arise. Without breakdowns, it is difficult to assess the mechanism’s effectiveness.
>
> **R4:** Thank you for raising the concerns regarding the timeline breakdown. To provide a clearer explanation, we provide a detailed per-step latency analysis across different RAG methods on PopQA dataset. Specifically, we treat the average **number of generated chunks** per answer as the number of **steps**, and we measure the time spent by each component of each RAG method during generation. We then divide the total time by the number of steps to obtain the average per-step latency for each component.
>
> For methods such as Self-RAG and SPS, where **multiple retrieval rounds** may occur within a single generation process, we record the total retrieval time across all rounds and divide it by the number of steps to compute the per-step retrieval cost.
> We fix the generation model to *Mistral-7B* for all methods. For Speculative RAG, we use *Mixtral-8×7B* as the verify model. The results are shown below:
>
> **All time reported in seconds (s).**
> |Method|steps|per-step generation time|per-step retrieve time|per-step total time|end to end time|
> |---|---|---|---|---|---|
> |Standard RAG|1.80|1.28|1.36|2.95|5.31|
> |Self-RAG|1.85|2.92|2.26|4.55|8.42|
> |Speculative RAG|2.26|4.11|1.32|5.43|12.28|
> |SPS|2.03|1.20|2.68|2.68|5.45|
>
> For each generation step, Self-RAG performs retrieval and generation *sequentially*, so its per-step latency is essentially the **sum of retrieval time and generation time** (as also illustrated in Figure 1). Spec.RAG incurs the **highest generation cost** because it performs multi-perspective sampling and relies on a larger verifier LLM, making each step substantially slower.
>
> In contrast, SPS achieves the **lowest** per-step generation time, since each parallel path operates on a much shorter context. Although SPS may trigger multiple retrieval rounds, its self-consistency selection is lightweight, and parallelism ensures that the per-step total latency becomes $$
> \text{per-step time} = \text{max} (\text{generation},\ \text{retrieve}),$$
> rather than their sum. This allows SPS to maintain inference time comparable to Standard RAG, while delivering higher accuracy.
>
>
> **References:**
>
> [1] Wenqi Jiang, et al. "Piperag: Fast retrieval-augmented generation via algorithm-system co-design." KDD, 2025.
>
> [2] Sebastian Borgeau, et al. "Improving language models by retrieving from trillions of tokens." In International Conference on Machine Learning, 2021.
>
> [3] Zilong Wang, et al. "Speculative rag: Enhancing retrieval augmented generation through drafting." International Conference on Learning Representations, 2024.
>
> [4] Akari Asai, et al. Self-RAG: Learning to retrieve, generate, and critique through self-reflection. In The Twelfth International Conference on Learning Representations, 2024.

---

> ### Author Response · Authors · 2025-11-27
>
> Dear Reviewer FtgG,
>
> Thank you very much for your thoughtful and detailed feedback. We have carefully addressed all the concerns you raised, including (i) providing more comprehensive comparisons with PipelineRAG and SpecRAG, along with additional experiments of PipeRAG conducted under identical retrievers and LLMs ; (ii) addressing the concern regarding missing ablation studies ; (iii) extending the experiments to larger LLMs and stronger retrievers; and (iv) providing a full latency decomposition to more clearly illustrate where the efficiency improvements arise.
>
> Your comments were extremely valuable and greatly improved the clarity and completeness of the paper. If any part of the updated experiments or analysis would benefit from further elaboration, we would be more than happy to clarify.
> Thank you again for your time and constructive suggestions.
>
> Best regards,
>
> All authors

---

### Official Review · Reviewer_DhGh · 2025-11-03

**Soundness:** 3
**Presentation:** 3
**Contribution:** 3
**Rating:** 6
**Confidence:** 4

**Summary:**

The paper introduces Staged Parallel Speculation (SPS), a training-free framework designed to improve the efficiency of Retrieval-Augmented Generation (RAG). SPS addresses the latency inherent in standard RAG systems by decoupling the retrieval and inference processes, allowing them to run in parallel. The framework employs a staged approach where retrieval for future text chunks occurs concurrently with the generation of current chunks.

Furthermore, SPS utilizes multi-perspective sampling to cluster retrieved documents into diverse subsets. It then generates multiple candidate answer chunks in parallel using these subsets. A key component is the training-free self-consistency selection mechanism, which identifies the most reliable candidate chunk based on semantic similarity to other candidates without requiring a trained verifier model. Experiments across five benchmarks (PopQA, TriviaQA, PubHealth, ARC-Challenge, ALCE-ASQA) demonstrate that SPS can achieve higher accuracy than standard RAG while offering significantly lower latency than other training-free speedup methods like Speculative RAG.

**Strengths:**

- The proposed method is training-free, making it a "plug-and-play" solution that can be easily adopted into existing RAG pipelines without the expensive and complex process of fine-tuning specialized drafter or verifier models.

- SPS demonstrates a compelling trade-off between latency and accuracy.

- Effectively combining staged parallel retrieval with a training-free self-consistency verifier for RAG chunks is an effective approach. The ablation studies (Table 3) validate the contribution of each component (staged retrieval, self-consistency, and multi-perspective sampling).

- The paper is well-organized. Figure 1 provides a clear visual contrast between sequential Self-RAG and parallel SPS , and Algorithm 1 succinctly describes the operational flow.

**Weaknesses:**

- Computational overhead. The claims regarding "efficiency" primarily refer to wall-clock latency, achieved through massive parallelization, not necessarily computational efficiency. For example, the experimental setup uses $m=5$ document subsets, meaning SPS requires launching 5 parallel endpoints of the generation model for every single query.

- Given that SPS uses 5x the inference compute (parallel streams) of Standard RAG, the comparison might be considered unfair in terms of total resources. For example, a more rigorous baseline would be "Standard RAG + Self-Consistency," where Standard RAG is also allowed to generate 5 paths (perhaps with temperature sampling or different retrieval subsets) and use the same voting mechanism.

- In abstract, the authors mention "Extensive experiments across multiple benchmark datasets show that SPS consistently surpasses training-free RAG baselines by achieving higher accuracy with 57% lower latency" - however, in Table 2, SPS's latency is similar to standard RAG. To make the claim more accurate, I would suggest using expressions like "... **at most** 57% lower latency".

**Questions:**

See weaknesses.

---

> ### Author Response · Authors · 2025-11-21
>
> We really appreciate your positive and instructive feekback and will provide detailed responses below.
>
> > **W1:** Computational overhead. The claims regarding "efficiency" primarily refer to wall-clock latency, achieved through massive parallelization, not necessarily computational efficiency. For example, the experimental setup uses m=5  document subsets, meaning SPS requires launching 5 parallel endpoints of the generation model for every single query.
>
> **R1:** Thank you for highlighting this point. We acknowledge that parallelization plays an important role in our latency improvements. This is an **inherent trade-off** of our design: we intentionally allocate more computation to achieve **substantially lower latency** (via parallelization) and higher answer accuracy.
>
> Although using m=5 subsets introduces a theoretical upper bound of **up to** 5× the compute of standard RAG, the practical overhead is **notably smaller** because each parallel path processes far fewer retrieved documents, leading to much shorter prompts and reduced per-path workload.
>
> Regarding memory usage, under **batch inference**, only a single model instance is required; the additional memory primarily comes from caching and storing hidden states for parallel paths. While the worst-case upper bound could approach 5× when context lengths are extremely long, in practice the context is much shorter, and the observed memory usage is only about 1–2× that of the standard RAG baseline.
>
>
>
> > **W2:** Given that SPS uses 5x the inference compute (parallel streams) of Standard RAG, the comparison might be considered unfair in terms of total resources. For example, a more rigorous baseline would be "Standard RAG + Self-Consistency," where Standard RAG is also allowed to generate 5 paths (perhaps with temperature sampling or different retrieval subsets) and use the same voting mechanism.
>
> **R2:** Thank you for raising this concern regarding the fairness of the comparison. We agree that Standard RAG + Self-Consistency constitutes a relevant and meaningful baseline. We have added new experiments to compare with SPS using *Mistral-7B* as backbone LLM, and report the resulting performance and latency below. Concretely, we set the sampling temperature to 0.8, incorporated the top-10 retrieved documents into the prompt, generated 5 samples, and applied the same self-consistency selection strategy used in SPS.
>
> Performance: accuracy(%)
> |Method|PopQA|TQA|Pub|ARC|
> |---|---:|---:|---:|---:|
> |Standard RAG|32.59|53.50|35.26|43.51|
> |Standard RAG+Self-Consistency|38.97|59.67|37.69|50.94|
> |SPS|50.11|68.12|64.35|71.76|
>
> Latency: time(seconds)
> |Method|PopQA|TQA|Pub|ARC|
> |---|---:|---:|---:|---:|
> |Standard RAG|5.36|5.15|5.62|5.09|
> |Standard RAG+Self-Consistency|5.45|5.31|5.67|5.64|
> |SPS|5.21|5.18|5.44|5.58|
>
>
> We observe that although applying self-consistency on top of Standard RAG indeed improves accuracy with only a slight increase in latency, our method still consistently achieves **higher accuracy** than Standard RAG + Self-Consistency. The superior performance of SPS comes from the staged retrieval design. Instead of performing a single retrieval step before generation like Standard RAG, retrieval in SPS is periodically triggered during the generation process. This approach ensures that the retrieved documents remain relevant to the latest context of the generation, allowing the LLM to better handle topic shifts that often occur when generating long answers to complex questions.
>
> Our latency is comparable to that of Standard RAG + Self-Consistency because, although SPS involves a more complex process for generating and selecting each chunk, the **multi-perspective sampling** significantly **shortens the context length** of each parallel path. These two effects offset each other, resulting in an overall latency that remains similar.
>
>
>
> > **W3:** In abstract, the authors mention "Extensive experiments across multiple benchmark datasets show that SPS consistently surpasses training-free RAG baselines by achieving higher accuracy with 57% lower latency" - however, in Table 2, SPS's latency is similar to standard RAG. To make the claim more accurate, I would suggest using expressions like "... at most 57% lower latency".
>
> **R3:** Thank you for pointing out this issue. The phrasing in our abstract was indeed somewhat misleading, and we agree with your suggestion that it should explicitly include “at most.” We will revise the wording accordingly in the revised manuscript. We sincerely appreciate your valuable feedback.

---

> ### Author Response · Authors · 2025-11-27
>
> Dear Reviewer DhGh,
>
> We would like to express our sincere gratitude for your thoughtful review and valuable feedback. We believe we have carefully addressed all the questions and concerns raised in your comments in our responses, and we hope our clarifications help resolve the issues you pointed out.
>
> If you have any further questions or would like additional clarification on any point, we would be very glad to respond promptly. Thank you again for your time and consideration.
>
> Best regards,
>
> All authors

---

### Meta-Review · Area_Chair_iUNy · 2026-01-03

**Summary:**

The paper proposes SPS, a training-free framework for RAG. The core contribution is a pipeline that parallelizes retrieval and generation steps and utilizes multi-perspective sampling with a self-consistency mechanism to select the best answer chunks. The authors claim this results in up to 57% lower latency compared to baselines while maintaining competitive accuracy.

**Reviewer Concerns:**

- Reviewers (DhGh, FtgG, brDz) appreciated the training-free nature of the approach, making it a "plug-and-play" solution compatible with existing LLMs without the need for complex fine-tuning of drafter/verifier models.

- The method demonstrates a clear reduction in wall-clock latency compared to sequential RAG approaches.

- The authors provided extensive experiments across multiple benchmarks (PopQA, TriviaQA, etc.) and offered additional ablation studies and breakdown analyses during the rebuttal phase.

**Reviewer Scores:**

- Reviewer FtgG pointed out that the core concepts—parallelizing retrieval (similar to PipeRAG) and speculative drafting (similar to Speculative RAG)—are well developed. While SPS adapts these for a training-free LLM setting, the architectural innovation is viewed as somewhat incremental. The differentiation relies heavily on the specific combination of existing techniques rather than a fundamental breakthrough.

- Reviewer brDz noted that while the method performs well for a training-free approach, it still lags behind fine-tuning-based methods in accuracy. Given that SPS incurs higher computational costs than standard inference, the value proposition—paying more compute for lower latency but sub-SOTA accuracy—is a niche trade-off that may not be compelling for a broad audience.

The paper presents an interesting engineering solution for reducing RAG latency. However, the ambiguity surrounding efficiency and the incremental nature of the contribution compared to existing parallel/speculative RAG works suggest the paper needs reframing or a stronger demonstration of resource-efficiency before publication

---

### Decision · Program_Chairs · 2026-01-26

Reject